# Evaluation of SARS-CoV-2-Positive Patients with Suspected Reinfection

**DOI:** 10.3390/v15112222

**Published:** 2023-11-07

**Authors:** Aytaj Allahverdiyeva, Ali Ağaçfidan, Lerzan Dogan, Mustafa Önel, Hayriye Kırkoyun Uysal, Alpay Medetalibeyoğlu, Naci Şenkal, Elvin Alaskarov, Sevim Meşe

**Affiliations:** 1Institute of Health Sciences, Istanbul University, Istanbul 34126, Turkey; lredzheb@gmail.com; 2Department of Medical Microbiology, Azerbaijan Medical University, Baku 370022, Azerbaijan; 3Department of Medical Microbiology, Istanbul Faculty of Medicine, Istanbul University, Istanbul 34093, Turkey; ali.agacfidan@istanbul.edu.tr (A.A.); onelm@istanbul.edu.tr (M.Ö.); hayriye.kirkoyun@istanbul.edu.tr (H.K.U.); 4Department of Internal Medicine, Istanbul Faculty of Medicine, Istanbul University, Istanbul 34093, Turkey; alpay.m@istanbul.edu.tr (A.M.); naci.senkal@istanbul.edu.tr (N.Ş.); 5Department of Otorhinolaryngology, Istanbul Medipol University, Istanbul 34230, Turkey

**Keywords:** COVID-19, SARS-CoV-2, RT-PCR, reinfection

## Abstract

The aim of this study was to investigate the reinfection rates and characteristics of SARS-CoV-2 in individuals with SARS-CoV-2 RNA present in their clinical specimens for COVID-19. Our data from the COVID-19 Laboratory of Istanbul University were analyzed for 27,240 cases between 27 March 2020 to 8 February 2022. Demographic characteristics, vaccination statuses, comorbidities, and laboratory findings were evaluated in cases with suspected reinfection, as determined by the presence of SARS-CoV-2 RNA at a rate of 0.3% in clinical specimens. When comparing laboratory values, leukocyte counts were lower in the second and third infections compared with the first infection (*p* = 0.035), and neutrophil counts were lower in the second infection (*p* = 0.009). Symptoms varied, with coughing being common in the first infection and malaise being common in subsequent infections. These results suggest that it is important to continue to monitor reinfection rates and develop strategies to prevent reinfection. Our results also suggest that clinicians should be aware of the possibility of reinfection and monitor patients for recurrent symptoms.

## 1. Introduction

Coronavirus disease 2019 (COVID-19) is known to have begun with the report of a cluster of pneumonia cases in people associated with the seafood and livestock market in Wuhan, China on 31 December 2019 [1,2] The disease, which progresses with severe symptoms including fever, shortness of breath, coughing, and weakness, quickly spread to other countries and became a global concern [3]. On 12 January 2020, as a result of sequencing samples from the cluster cases, Chinese authorities declared that the virus causing the disease was a new member of the Coronaviridae family, similar to SARS-CoV-2 (severe acute respiratory syndrome coronavirus) and MERS-CoV (Middle East respiratory syndrome coronavirus). On 11 February 2020, it was named SARS-CoV-2 by the International Committee on Taxonomy of Viruses (ICTV) [2,4]. SARS-CoV-2 enters the cell by binding to the angiotensin-converting enzyme (ACE-2) receptor via spike (S) proteins on the virion surface [4,5,6]. The S protein is one of the most important factors involved in the generation of the immune response against SARS-CoV-2, including neutralizing antibodies [7]. Therefore, almost all vaccines developed against SARS-CoV-2 are based on the S protein [8,9,10].

Genome sequencing, previous mRNA vaccine candidates, adenovirus vectors, etc., have been instrumental in the successful development of COVID-19 vaccines [11]. To accelerate the development and production of COVID-19 vaccines and to ensure fair and equitable access worldwide, a large multilateral fund called COVAX was established by the WHO, Gavi, UNICEF, and the Coalition for Epidemic Preparedness Innovations (CEPI). COVAX was initially focused on providing 2 billion doses by the end of 2021, with the goal of vaccinating 20% of the population and ensuring equitable distribution so that the world’s most vulnerable people were vaccinated first [12]. The first administration of COVID-19 vaccine in Turkey took place on 13 January 2021. The first phase of vaccination specifically targeted healthcare workers, adults, especially those at risk, and the elderly. Persons younger than 18 years were not eligible for vaccination. The administration of booster shots was subject to strict time frames depending on the type of vaccine and the age of the patient. Booster intervals ranged from 3 to 12 weeks. Various campaigns were conducted in Turkey through television and social media to encourage the public to receive the COVID-19 vaccine. Public figures were vaccinated in live broadcasts [13].

One of the most important questions in predicting the course of the COVID-19 pandemic is the duration of immune responses that provide protection against reinfection [14]. For some viruses, the initial infection can provide lifelong immunity. However, cases of reinfection with SARS-CoV-2 are being reported with increasing frequency [15,16]. As defined by the Center for Disease Control and Prevention, “reinfection” refers to the recurrence of a completely resolved infection caused by a pathogen after a period of time [17]. For individuals with symptoms similar to COVID-19, reinfection is defined as a repeat positive PCR test 45–89 days after the initial SARS-CoV-2 infection. For cases of SARS-CoV-2 infection in the absence of COVID-19-like symptoms, “reinfection cases” are those with a positive PCR test 90 days after the initial infection.

The aim of this study was to investigate reinfection rates and characteristics according to CDC criteria in individuals with SARS-CoV-2 RNA in a clinical sample. For this purpose, we evaluated the demographic and clinical characteristics, vaccination status, and laboratory findings of the suspected reinfection cases by examining the data of the COVID-19 Laboratory of Istanbul University, Faculty of Medicine, which receives patients from all regions of Turkey and different communities. The results of our study are expected to contribute to the planning of protective measures and treatment procedures against COVID-19.

## 2. Materials and Methods

### 2.1. Study Group

Considering the CDC reinfection criteria, the inclusion and exclusion criteria for this study are presented in Figure 1. In this study, we identified patients with positive SARS-CoV-2 qPCR test results, along with the time between the first positive test and the second positive test based on the criteria established by the CDC. Included in this study as “reinfection cases” were individuals who exhibited symptoms similar to COVID-19 and who tested positive via PCR at 45–89 days since their initial SARS-CoV-2 infection, and those who tested positive for PCR after 90 days since their first SARS-CoV-2 infection but did not exhibit COVID-19-like symptoms. Latent cases that remained positive for a long time were not included in the study.

### 2.2. SARS-CoV-2 RNA Test

SARS-CoV-2 RNA test results from 27 March 2020 to 8 February 2022 were reviewed at the COVID-19 Laboratory of the Department of Virology and Basic Immunology, Istanbul University Faculty of Medicine. The SARS-CoV-2 RNA test was performed in this laboratory on combined oropharyngeal and nasopharyngeal swab specimens. The test was performed using vNAT (viral nucleic acid isolation) and transcriptase (RT) real-time (q) PCR kits manufactured by Bioeksen (Sarıyer, Turkey), and provided by the Ministry of Health. The viral nucleic acid isolation procedure was performed manually in the BGD-2 cabinet. RT-qPCR was performed on the Rotor Gene Q (QIAGEN, Hilden, Germany). It was performed using a one-step RT-qPCR kit targeting the specific N and Orf1ab gene regions of SARS-CoV-2. The human RNaseP gene was targeted for nucleic acid extraction and inhibition control.

The test results were interpreted according to the procedure of the Bio-speedy kit (Bioeksen, Turkey) and amplification curves were examined. If a Ct value was assigned to a sample by the instrument software and the curve was sigmoidal, the Ct value could be used in the final evaluation. Non-sigmoidal curves were recorded as negative. If a Ct value was assigned to a sample but the curve was not sigmoidal, the result was recorded as negative.

If the SARS-CoV-2 Cq FAM value was ≤30, it was considered to signify the presence of SARS-CoV-2 RNA in a clinical sample.

If the SARS-CoV-2 Cq FAM was greater than 30, it was not considered to signify the presence of SARS-CoV-2 RNA in a clinical sample.

The Ct-HEX(IC) value was examined in individuals who showed a suspicious sigmoid curve below the threshold in the FAM channel.

### 2.3. Data Collection

The retrospective information of the individuals was retrieved from the Public Health Management System (PHMS) and the Laboratory Information Management System (LIMS). The laboratory findings C-reactive protein (CRP), white blood cell (WBC), neutrophil (Neu) count, lymphocyte count, ferritin level, D-dimer level, hemoglobin level (Hgb), blood urea nitrogen level (BUN), and clinical symptoms of chronic disease were evaluated. The data were transferred to a software program (Excel 2021, Microsoft, Redmond, WA, USA), and statistical analyses were conducted.

### 2.4. Statistical Analysis and the Ethics Committee

Statistical analyses were conducted using the SPSS 21 software package. The normality of continuous data was evaluated via the Kolmogorov–Smirnov and Shapiro–Wilk tests. Continuous data were presented as mean ± standard deviation if normally distributed, and as median (minimum–maximum) if not normally distributed. The Friedman test was used, which is appropriate for comparing two dependent groups using the dependent-groups *t*-test and Wilcoxon test. This test was chosen because the data did not follow a normal distribution when comparing the three groups. Chi-square and Fisher’s exact tests were used to compare categorical data. The Mann–Whitney U test was used to compare two independent groups.

The factors influencing whether the Ct value was 25–30 or below 25 were assessed using binomial logistic regression analysis. The fit of the model was examined using the Hosmer–Lemeshow test. All tests were bilateral, and a significance level of *p* < 0.05 was considered.

This retrospective study was approved by the Istanbul University Istanbul Medical Faculty Clinical Research Ethics Committee (decision number: 2022/1579, dated 23 September 2022), and the study adhered to the principles of the Declaration of Helsinki at all stages.

## 3. Results

### 3.1. Background Demographics

After evaluating the positive SARS-CoV-2 RNA test results (27,240 cases) from 27 March 2020 to 8 February 2022 at the COVID-19 Laboratory of the Istanbul University Medicine Faculty, we identified 82 (0.3%) patients who met the CDC criteria for reinfection.

Of the 82 patients included in the study, 54.9% were female and 45.1% were male. The mean age of the males was 34.81 ± 12.97 years, while the mean age of the females was 35.38 ± 14.92 years. The overall mean age of all patients was 35.1 ± 13.9 years (range: 18–81).

Demographic data of the patients are given in Table 1.

### 3.2. Comparison of the Clinical Characteristics of the Patients

The patient population, believed to include 82 reinfections, consisted of 37 individuals (45.1%) without chronic diseases. Of those with chronic diseases (54.9%), the breakdown was as follows: 20 (24.4%) with allergies, 9 (11%) with oncology-related conditions, 4 (4.9%) with diabetes mellitus, 4 (4.9%) with hypertension, 3 (3.7%) with rheumatology-related conditions, and 2 (2.4%) with thyroid diseases (Table 2).

Coughing was the most prevalent symptom in the initial infection, while fatigue was the most common symptom in the second and third reinfections (Figure 2).

### 3.3. Comparison of Laboratory Values of the Patients

When comparing the laboratory values of patients with initial, second, and third recurrent infections, it was found that the WBC values in the second and third infections were significantly lower than those in the initial infection (*p* = 0.035) (Table 3). Additionally, the Neu value was found to be significantly lower in the second infection compared to the initial infection (*p* = 0.009). No difference was found between the other laboratory values included in the study (Table 3).

We divided the patients into three groups based on the PCR in-cycle threshold (Ct): those with high, moderate, or low Ct values (Ct < 25, 25–30, or >30, respectively). The low Ct value was not detected in our study. Based on this, we classified the patients into two groups: moderate and high Ct values. There was no significant difference between Ct values, gender, age, order of infection, vaccination status, and vaccine type (*p* = 0.659, *p* = 0.903, *p* = 1.00, *p* = 0.108, *p* = 1.00, respectively).

The Turkish public exhibited reluctance to accept the COVID-19 vaccine. For future vaccination campaigns, disseminating the scientific stance on novel coronavirus mutations to a wider audience could potentially alleviate vaccine hesitancy. During this study, two types of COVID-19 vaccines were distributed in Turkey: BNT162b2 and CoronaVac.

In our study, 41.89% (31/74) of secondary reinfection cases were not vaccinated, while 41.89% (31/74) were vaccinated with BNT162b2, and 14.86% (11/74) were vaccinated with CoronaVac. A second reinfection case was vaccinated with both. In cases of third reinfection, these rates were 50% (4/8), 37.5% (3/8), and 12.5% (1/8), respectively. When comparing second and third infections based on vaccination type and no vaccination, we observed that there was no significant correlation between vaccines and reinfection status (*p* = 0.919) (Table 4).

## 4. Discussion

The highly variable genetics of SARS-CoV-2 and the emergence of new variants have been considered as the main cause of reinfections [18]. Reinfection may also occur due to host-related reasons, such as the decrease over time in the protective immunity developed against SARS-CoV-2 through natural infection or vaccination. However, there is a need to investigate reinfections in terms of epidemiological, physio-pathological and clinical features [19,20,21]. In our study, we examined the demographic, clinical, and laboratory characteristics of patients with suspected SARS-CoV-2 reinfection, which we discerned from Istanbul Medical Faculty Virology Laboratory data according to CDC criteria.

According to our results, the rate of patients with suspected SARS-CoV-2 reinfection was determined to be 0.35%, of which 54.9% were female and 45.1% were male patients. In our study, we found that the mean age of patients with suspected reinfection was 35.1 ± 13.9 years, and 54.9% of them had a chronic disease. In a study conducted in Iran between 20 March 2020 and 20 November 2020 (9 months), the incidence of reinfection was estimated to be 2.5 per thousand patients, that is, 1.98 per thousand in females and 2.96 per thousand in males. In this study, there was no statistically significant difference in the risk of reinfection between men and women [22]. Although demographics vary from country to country, incidence rates appear to be similar. A study from Qatar, in which swab PCR-positive symptomatic cases 45 days after initial infection were considered as reinfection, suggested that the estimated risk of reinfection was 0.2% [23]. Similar to our study, in a Malawi study covering the period 2020–2022, it was found that 55.76% of reinfection cases detected at a rate of 0.18% with an interval of at least 90 days were male and 44.24% were female [20].

Lu et al., in their study, found that out of the 87 cases with repeat positivity, 46 of them had mild clinical symptoms and 41 cases had moderate clinical symptoms during their first hospitalization. In the same study, it was observed that out of the 87 cases that tested positive again after discharge, 77 were asymptomatic. However, 10 of the cases did exhibit mild coughing symptoms, primarily at night. Severe clinical findings were not observed in these re-positive cases [24].

From a clinical point of view, we found that 60% of the cases with suspected reinfection were symptomatic. Similarly, in Malawi, the rate of symptomatic cases was determined to be 65.17% [21]. In the Qatar study, twenty-three reinfection cases (42.6%) were diagnosed in a healthcare facility, which was attributed to the fact that the patients may have been symptomatic. The authors explained that 31 reinfection cases (57.4%) were asymptomatic cases detected incidentally through random testing campaigns/surveys or contact tracing. In our study, which only included symptomatic cases, coughing was the most common symptom in the first infection, while fatigue was the most common in the second infection (29.3%) followed by the third infection (24.40%) and the fourth infection (54.50%). The least common symptoms were headache, which occurred in 7.30% of cases during the first infection, loss of taste and smell, which occurred in 4.90% of cases during the second infection, and headache, which did not occur at all during the third infection. In a study examining 209 reinfection cases, it was determined that 121 patients were symptomatic. In this study, the prevalence levels of various symptoms were generally similar between the primary and secondary infections. The exception to this was diarrhea, which was significantly more common in the primary infection as compared to the secondary infection [25]. In another study by Tan et al., no significant difference was found between primary and reinfection in the proportion of individuals with fever, acute respiratory symptoms, or who were asymptomatic [21].

We also evaluated the laboratory values of initial, second, and third recurrent infections separately. We found that the WBC values in the second and third infections were lower than in the first infection. Additionally, we observed that the Neu values were significantly lower in the second infection compared to the initial infection. The first reinfection case confirmed by genomic sequencing was reported from Hong Kong, with mild symptoms and normal laboratory findings except for elevated CRP and hypokalemia [26]. The second confirmed case, later reported from Belgium, was a 57-year-old asthma patient who was immunocompetent but was receiving daily doses of inhaled steroids. It was reported that this patient did not show any abnormalities in blood count and routine biochemistry except for a slight increase in liver enzymes. The patient, whose oxygen saturation was determined as 94%, showed improvement with long-term rest without being hospitalized [27]. On the other hand, increased creatinine levels and WBC count have been associated with mortality in hospitalized reinfection cases [28]. It is crucial to scrutinize the factors that could impact the clinical course and severity of COVID-19 reinfections, particularly in individuals with chronic illnesses. The differences in laboratory parameters we detected in reinfection cases demonstrated the need for more comprehensive research and case studies in this field. Through more comprehensive studies, we can better understand the underlying mechanisms and risk factors associated with reinfection. These studies can provide the data necessary to develop strategies for the prevention, early detection, and management of reinfections, eventually contributing to improving patient care and public health interventions. Although our experimental data are somewhat limited, our ongoing research in this field will provide valuable insights into the long-term effects of COVID-19 and steer efforts to decrease the risks linked to reinfection.

In our study, when we evaluated the SARS-CoV-2 RNA test results according to the initial, second, and third infection, we found the mean Ct values were 17.58 ± 3.55, 17.95 ± 3.62, and 16.66 ± 2.13, respectively. In a study conducted in Singapore, the Ct value of PCR at the second episode was found to be significantly lower in reinfection cases (mean 23; 95%—CI 20–26) as compared to non-reinfection cases (mean 34; 95%—CI 32–36) [28]. Yonatan et al. suggest that certain precautions should be taken when interpreting the Ct data in their study conducted in Israel. They mention that while PCR efficiency is likely comparable for the Delta and Omicron variants, it is advised to avoid comparing Ct values between variants and instead focus on intravariant comparisons [25].

In our study, we evaluated the development of reinfection according to the BNT162b and CoronaVac vaccines. There was no significant association between reinfection and vaccine type. However, we believe that the small sample size may have influenced this statistical evaluation. A study conducted on a large group of healthcare workers in our country showed that the BNT162b vaccine induced higher levels of IgG in individuals with COVID-19 [2].

Better measurement of the relationship between chronic diseases and at-risk populations is essential for future health system planning. In our study, we found that allergic diseases were the most common comorbidity in suspected reinfection cases with a rate of 24.4%. Similar to our study, Nguyen et al. reported that 68% of reinfection patients had at least one comorbidity, of which chronic respiratory diseases were the most common (26%) [21]. In another study conducted in our country, hypertension ranked first with a rate of 34%, while asthma and allergic diseases ranked fourth with a rate of 24% [27]. These findings indicate that the role of chronic diseases in SARS-CoV-2 reinfection, especially respiratory diseases of allergic or non-allergic origin, should be investigated. Such information will also support policy decisions and allow consideration of the various economic, social, and health impacts of preventive interventions, including societal restrictions.

In cases of reinfection, it is possible that individuals who have acquired natural immunity and recovered from their initial infections may not develop neutralizing antibodies or protective immune responses strong enough to defend against a new infection. It is also possible that this ability may diminish over time [28]. One of the limitations of our study is that antibody levels were not measured during the initial, second, and third periods of infection with SARS-CoV-2. However, since it is not possible to make a definitive interpretation of the protective effects of antibodies without determining their neutralizing activities, routine measurement of antibody levels is not recommended for immune response monitoring. Therefore, despite immunization after natural infection or vaccination, it is important not to give up physical distancing, the use of masks, and hygiene rules to protect against reinfection [29]. It is crucial to examine the factors that could impact the clinical course and severity of COVID-19 reinfections, particularly in individuals with chronic illnesses. The observed distinctions in laboratory parameters in reinfection cases emphasize the importance of conducting extensive research and case studies within this domain. Through more thorough examinations, we can investigate the underlying mechanisms and risk factors linked with reinfection. Our expertise can inform methods to prevent, detect early, and manage reinfections, leading to improved patient care and public health interventions. Although our experimental data are somewhat limited, continuous research in this field will offer valuable insights into the long-term effects of COVID-19 and steer efforts to decrease the risks linked to reinfection.

Despite the advanced diagnostic techniques used today, diagnosing reinfection for COVID-19 remains challenging. Prolonged viral shedding, reactivation, relapse, test-related false positivity, and the presence of viral genomic fragments challenge the differential diagnosis of reinfection. Therefore, clinical, laboratory, and verification criteria are recommended for the differential diagnosis of COVID-19 reinfection. These criteria can be summarized as the recovery of the primary infection, the long-term interval between primer infection and reinfection, clinical and laboratory findings indicating acute infection, a high level of viral RNA (ribonucleic acid), decreased protective immunity, and verification of reinfection using genomic sequencing [19,20,28]. In this context, one of the limitations of our study is that cases with suspected reinfection were not confirmed using genomic sequencing due to a lack of funding. However, as the pandemic continues in our country, genomic surveillance is carried out at the National Virology Reference Laboratory affiliated with the Ministry of Health. Authorized laboratories, including our laboratory, send samples to this reference laboratory for genetic analysis on a weekly basis [29]. Genomic sequence results are reported to GSIAD by the reference laboratory. According to GSIAD data, VOC Alpa GRY (B.1.1.7 + Q*) appeared in Turkey between October 20,020 and February 2021. Later, VOC Alpa (B.1.1.7 + Q*), VOC Delta GK (B.1.617.2 +AY.*), VOC Beta GH/501Y.V2 (B.1.352 + B.1.351.2 + B.1.351.3), and VOC Gamma GR/501Y.V3 (P.1 + P.1*) were found mixed in circulation until June 2021. Then, until the end of 2021, the Delta variant dominated in Turkey. In 2022, the Omicron GRA (B.1.1.529 + BA.*) variant was found to be dominant in Turkey (Figure 3). As can be understood from these data, the circulation of another variant in Turkey from time to time increases the risk of reinfection.

Although we did not perform genomic sequencing in our study, we used inclusion and exclusion criteria that increased the likelihood of true reinfection when creating the study group. We included cases in which the interval between the first and second infection was at least 45 days for those with symptoms and at least 90 days for those without symptoms. We excluded the presence of SARS-CoV-2 RNA persisting for a long time without recovery. In addition, the fact that most of the patients in our study (73.2%) were symptomatic and had laboratory data indicating acute infection supports our findings. Despite these positive factors, we defined our study as cases with suspected reinfection because we could not confirm it with genomic sequencing. Furthermore, although the single-center nature of this retrospective study is considered a disadvantage, it allows a general evaluation with a broad patient profile from our hospital. However, due to the ever-changing variations in SARS-CoV-2, there is still a need to monitor reinfection with a larger number of cases.

## 5. Conclusions

On 5 May 2023, the World Health Organization declared the global health emergency to have ended. Nevertheless, SARS-CoV-2 continues to pose a threat as it spreads globally. Additionally, numerous individuals still experience non-negligible long-term effects of COVID-19, known as long COVID, while SARS-CoV-2 variants or subvariants are still emerging.

Continuous monitoring of reinfection cases involving a larger number of people is crucial due to the ever-evolving variations in SARS-CoV-2. Conducting larger-scale surveillance can provide researchers with a better understanding of reinfection dynamics and their relationship with different virus variants. Monitoring more reinfection cases can also yield insights into the efficacy of immunity from reinfections or vaccines against specific virus variants. Therefore, it is necessary to analyze more cases to identify potential differences in clinical presentation, laboratory values, and disease severity between the various strains. This will provide us with comprehensive and reliable data that can inform evidence-based decisions about public health interventions, including vaccination campaigns and containment strategies.

While 66% of the population in Turkey was fully vaccinated as of May 2022 [13], our study covering this period showed that reinfection occurred at a significant rate. However, comprehensive studies are needed to show how reinfection rates change with the new variants and the use of new vaccines. In our study, it was determined that most of the reinfection cases were symptomatic and associated with chronic diseases. Our study has shown that COVID-19 reinfections present with significant variation in laboratory parameters, and, therefore, factors affecting the clinical course and severity of reinfection are issues that require further research with more case studies.

## Figures and Tables

**Figure 1 viruses-15-02222-f001:**
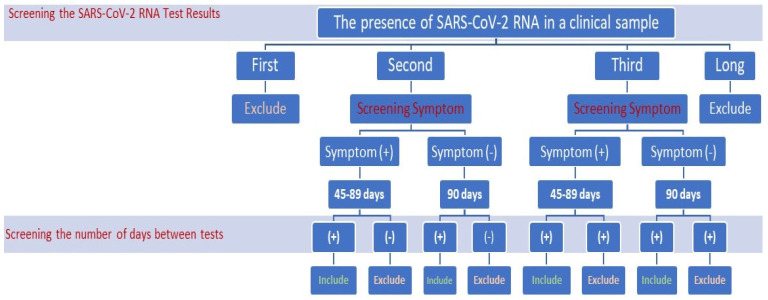
Diagram showing the study group.

**Figure 2 viruses-15-02222-f002:**
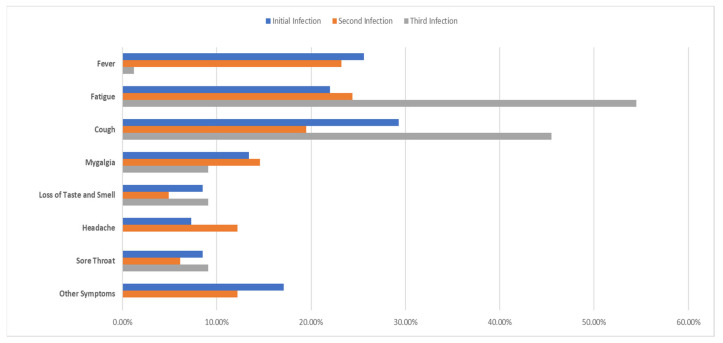
Prevalence of symptoms in infections.

**Figure 3 viruses-15-02222-f003:**
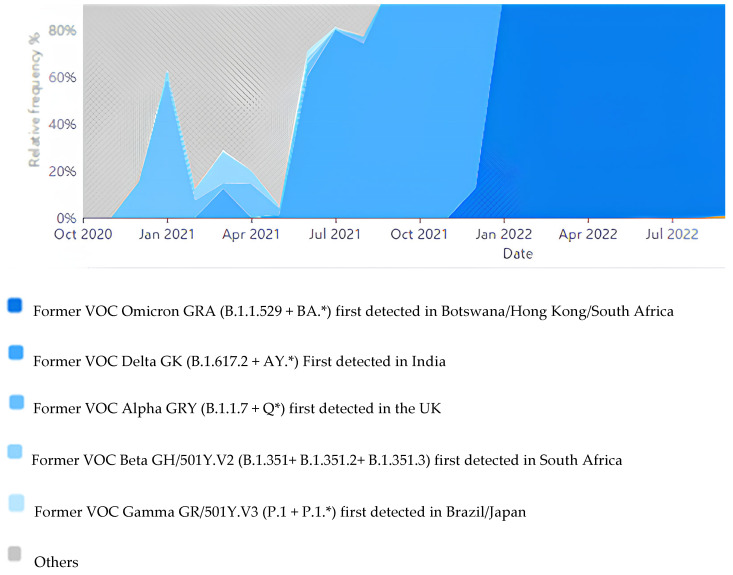
SARS-CoV-2 variants changing over time in Turkey (adapted from GISAID).

**Table 1 viruses-15-02222-t001:** Distribution of demographic data of participants.

Categories	*n* (%)	Mean ± Standard Deviation	Median (Min–Max)
Gender			
Female	45 (54.9%)		
Male	37 (45.1%)		
Age		35.1 ± 13.9	30.5 (18–81)
Have chronic diseases			
Yes	45 (54.9%)		
No	37 (45.1%)		
Recurrent infection			
Second	74 (90.2%)		
Third	8 (9.8%)		
Ct value			
Initial infection		17.58 ± 3.55	17.35 (11.03–27.70)
Second infection		17.95 ± 3.62	17.42 (10.34–28.95)
Third infection		16.66 ± 2.13	16.33 (12.86–22.60)

**Table 2 viruses-15-02222-t002:** Chronic disease state.

		*n*	%
Chronic Disease	None	37	45.1
	Allergic	20	24.4
	Oncological	9	11.0
	Diabetes	4	4.9
	Blood Pressure	4	4.9
	Rheumatological	3	3.7
	Cardiological	3	3.7
	Thyroid	2	2.4

**Table 3 viruses-15-02222-t003:** The comparison of laboratory values of initial, secondary, and third infections.

Categories	Initial Infection	Second Infection	Third Infection	*p * ^1^
	Median (Min–Max)	Median (Min–Max)	Median (Min–Max)	
WBC	7302 (1283–12,370)	6729 (4300–13,130)	5890 (1029–14,240)	**0.035 ^2^**
Neu	4910 (1743–8729)	3590 (1538–7130)	4528 (1500–9700)	**0.009 ^3^**
Lym	2029 (600–3300)	1200 (958–2391)	1002 (800–3100)	0.234
Bun	13.1 (7.3–27.4)	12.5 (6.7–22.5)	12.4 (7–27.6)	0.670
CRP	1.7 (0.2–102)	2.2 (0.2–79)	3 (0.1–80)	0.695
Ferritin	102 (20–1597)	103 (24–434)	110 (25.4–300)	0.643
D-dimer	210 (52–730)	221 (38–992)	189 (47–1485)	0.441

^1^ Friedman test, ^2^ 2nd and 3rd infection values are lower than 1st infection values, ^3^ 2nd infection values are lower than 1st infection values. WBC: white blood cells; Neu: neutrophils; Lym: lymphocytes; Bun: urea; CRP: C-reactive protein.

**Table 4 viruses-15-02222-t004:** Comparison of second and third infection by vaccine type.

		Vaccine			
		None	BNT162b2	CoronaVac	
		*n* (%)	*n* (%)	*n* (%)	*p*
Reinfection	Second *	31 (88.6)	31 (91.2)	11 (91.7)	0.919
	Third	4 (11.4)	3 (8.8)	1 (8.3)	0.919

*p* < 0.05 is considered significant, chi-squared. * A second reinfection case was vaccinated with both.

## Data Availability

The retrospective information of the individuals was retrieved from the Public Health Management System (PHMS) and the Laboratory Information Management System (LIMS). hsgm.saglik.gov.tr/tr/web-uygulamalarimiz/285-laboratuvar-bilgi-yönetim-sistemi.html, http://www.probel.com.tr/ accessed on 1 January 2020.

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
