# Peer review of "Evaluation of SARS-CoV-2-Positive Patients with Suspected Reinfection"

_viruses, 2023, doi:10.3390/v15112222_

Round 1

Reviewer 1 Report (Previous Reviewer 1)

Comments and Suggestions for Authors

The manuscript has been improved.

I still have some minor comments:

- figure 1 and 3: the lettering is very small and difficult to read. Please enlarge.

- Table 3: Format titles better: align min and max on one line

- Table 4: Align None-BNT-Coronavac on one line. Do not crop "second".

Author Response

Yes        Can be improved            Must be improved             Not applicable

Does the introduction provide sufficient background and include all relevant references?

(x)         ( )          ( )          ( )

Are all the cited references relevant to the research?

(x)         ( )          ( )          ( )

Is the research design appropriate?

(x)         ( )          ( )          ( )

Are the methods adequately described?

(x)         ( )          ( )          ( )

Are the results clearly presented?

( )          (x)         ( )          ( )

Are the conclusions supported by the results?

(x)         ( )          ( )          ( )

-The evaluation has been taken into account. Thanks.

Comments and Suggestions for Authors

The manuscript has been improved.

I still have some minor comments:

- figure 1 and 3: the lettering is very small and difficult to read. Please enlarge.

- Table 3: Format titles better: align min and max on one line

- Table 4: Align None-BNT-Coronavac on one line. Do not crop "second".

-Thank you for the correction. It's all done.

Reviewer 2 Report (New Reviewer)

Comments and Suggestions for Authors

The study aimed to investigate reinfection rates and characteristics of SARS-CoV-2 in individuals with the presence of SARS-CoV-2 RNA in their clinical specimens for COVID-19. The analysis revealed certain findings. Leukocyte counts were lower in the second and third infections compared to the first infection, and neutrophil counts were lower in the second infection. Symptoms varied among reinfections, with cough being common in the first infection and malaise being common in subsequent infections. These results highlight the importance of monitoring reinfection rates and developing strategies to prevent reinfection. Clinicians should be aware of the possibility of reinfection and closely monitor patients for recurrent symptoms. It is crucial to continue research on reinfection to better understand its dynamics and implications for public health. The topic is important and the manuscript provides an analysis of the subject. I recommend accepting this article after MINOR REVISIONS.

1.     It is indeed important to investigate the factors that may affect the clinical course and severity of COVID-19 reinfections, especially in individuals with chronic diseases. The variation in laboratory parameters observed in reinfection cases further emphasizes the need for more extensive research and case studies in this area. By conducting more comprehensive studies, researchers can gain a better understanding of the underlying mechanisms and risk factors associated with reinfection. This knowledge can help inform strategies for prevention, early detection, and management of reinfections, ultimately contributing to better patient care and public health interventions. Although the experimental data is somewhat limited, continued research in this field will provide valuable insights into the long-term impact of COVID-19 and guide efforts to mitigate the risks associated with reinfection. Consider discussing the limitations of your study and potential future directions for research.

2.     Check the abbreviations throughout the manuscript and introduce the abbreviation when the full word appears the first time. “SARS-CoV and MERS-CoV” in line 39.

3.     There is a lack of literature citations. The authors should enrich the related articles. For example, in page 1, “The S protein is one of the most important factors involved in the generation of the immune response against SARS-CoV-2, including neutralizing antibodies. (DOI: 10.3390/cells11030437)”; in lines 47-48, “Genome sequencing, previous mRNA vaccine candidates, adenovirus vectors, etc. have been instrumental in the successful development of COVID-19 vaccines. (DOI: 10.1016/j.csbj.2020.11.011)”; in lines 62-63, “One of the most important questions in predicting the course of the COVID-19 epidemic is the duration of immune responses that provide protection against reinfection. (DOI: 10.3390/nu15153443)”; in lines 228-229, “The highly variable genetics of SARS-CoV-2 and the emergence of new variants have been considered as the main cause of reinfections.(DOI: 10.3389/fimmu.2022.898192)”.

4.     In “5. Conclusions”, “Despite ongoing vaccination efforts globally, COVID-19 infections are still a public health concern.” Please make necessary corrections: “On 5 May 2023, the World Health Organization declared the end of the global health emergency. However, SARS-CoV-2 remains a threat as it continues to spread globally. In addition, many people also continue to suffer from non-negligible long-term effects of COVID-19 (long COVID), and SARS-CoV-2 variants or subvariantsare still emerging”

5.     With the constantly evolving variations of SARS-CoV-2, it is crucial to continuously monitor reinfection cases with a larger number of individuals. This will allow researchers to better understand the dynamics of reinfection and its relationship with different variants of the virus. Monitoring reinfection cases on a larger scale can provide insights into the effectiveness of immunity acquired from previous infections or vaccinations against specific variants. It can also help identify any potential differences in clinical manifestations, laboratory parameters, and disease severity associated with different variants. By tracking and analyzing a larger number of cases, researchers can gather more robust data that can contribute to evidence-based decision-making in public health measures, such as vaccination strategies and containment measures.

Comments on the Quality of English Language

Minor editing

Author Response

Yes        Can be improved            Must be improved             Not applicable

Does the introduction provide sufficient background and include all relevant references?

( )          (x)         ( )          ( )

Are all the cited references relevant to the research?

( )          (x)         ( )          ( )

Is the research design appropriate?

( )          (x)         ( )          ( )

Are the methods adequately described?

( )          (x)         ( )          ( )

Are the results clearly presented?

( )          (x)         ( )          ( )

Are the conclusions supported by the results?

( )          (x)         ( )          ( )

-The evaluation has been taken into account. Thanks.

Comments and Suggestions for Authors

The study aimed to investigate reinfection rates and characteristics of SARS-CoV-2 in individuals with the presence of SARS-CoV-2 RNA in their clinical specimens for COVID-19. The analysis revealed certain findings. Leukocyte counts were lower in the second and third infections compared to the first infection, and neutrophil counts were lower in the second infection. Symptoms varied among reinfections, with cough being common in the first infection and malaise being common in subsequent infections. These results highlight the importance of monitoring reinfection rates and developing strategies to prevent reinfection. Clinicians should be aware of the possibility of reinfection and closely monitor patients for recurrent symptoms. It is crucial to continue research on reinfection to better understand its dynamics and implications for public health. The topic is important and the manuscript provides an analysis of the subject. I recommend accepting this article after MINOR REVISIONS.

  1. It is indeed important to investigate the factors that may affect the clinical course and severity of COVID-19 reinfections, especially in individuals with chronic diseases. The variation in laboratory parameters observed in reinfection cases further emphasizes the need for more extensive research and case studies in this area. By conducting more comprehensive studies, researchers can gain a better understanding of the underlying mechanisms and risk factors associated with reinfection. This knowledge can help inform strategies for prevention, early detection, and management of reinfections, ultimately contributing to better patient care and public health interventions. Although the experimental data is somewhat limited, continued research in this field will provide valuable insights into the long-term impact of COVID-19 and guide efforts to mitigate the risks associated with reinfection. Consider discussing the limitations of your study and potential future directions for research.

-Thank you for your evaluation, we are very pleased. Based on your comment, we have discussed our limitations, especially in individuals with chronic diseases, in the relevant section (Line 289-298) as follows.

It is crucial to scrutinize the factors that could impact the clinical course and severity of COVID-19 reinfections, particularly in individuals with chronic illnesses. The differences in laboratory parameters we detected in reinfection cases demonstrated the need for more comprehensive research and case studies in this field. Through more comprehensive studies, we can better understand the underlying mechanisms and risk factors associated with reinfection. These studies can provide the data necessary to develop strategies for the prevention, early detection and management of reinfections, eventually contributing to improving patient care and public health interventions. Although our experimental data is somewhat limited, our ongoing research in this field will provide valuable insights into the long-term effects of COVID-19 and steer efforts to decrease the risks linked to reinfection.

  1. Check the abbreviations throughout the manuscript and introduce the abbreviation when the full word appears the first time. “SARS-CoV and MERS-CoV” in line 39.

-Thank you for your warning. The abbreviation has been reviewed.

  1. There is a lack of literature citations. The authors should enrich the related articles. For example, in page 1, “The S protein is one of the most important factors involved in the generation of the immune response against SARS-CoV-2, including neutralizing antibodies. (DOI: 10.3390/cells11030437)”; in lines 47-48, “Genome sequencing, previous mRNA vaccine candidates, adenovirus vectors, etc. have been instrumental in the successful development of COVID-19 vaccines. (DOI: 10.1016/j.csbj.2020.11.011)”; in lines 62-63, “One of the most important questions in predicting the course of the COVID-19 epidemic is the duration of immune responses that provide protection against reinfection. (DOI: 10.3390/nu15153443)”; in lines 228-229, “The highly variable genetics of SARS-CoV-2 and the emergence of new variants have been considered as the main cause of reinfections.(DOI: 10.3389/fimmu.2022.898192)”.

-References have been added. Thanks for you contribution.

  1. In “5. Conclusions”, “Despite ongoing vaccination efforts globally, COVID-19 infections are still a public health concern.” Please make necessary corrections: “On 5 May 2023, the World Health Organization declared the end of the global health emergency. However, SARS-CoV-2 remains a threat as it continues to spread globally. In addition, many people also continue to suffer from non-negligible long-term effects of COVID-19 (long COVID), and SARS-CoV-2 variants or subvariantsare still emerging”

-Your point has been added. Thanks for your contribution.

On May 5, 2023, the World Health Organization declared the global health emergency to have ended. Nevertheless, SARS-CoV-2 continues to pose a threat as it spreads globally. Additionally, numerous individuals still experience non-negligible long-term effects of COVID-19, known as long COVID, while SARS-CoV-2 variants or subvariants are still emerging.

  1. With the constantly evolving variations of SARS-CoV-2, it is crucial to continuously monitor reinfection cases with a larger number of individuals. This will allow researchers to better understand the dynamics of reinfection and its relationship with different variants of the virus. Monitoring reinfection cases on a larger scale can provide insights into the effectiveness of immunity acquired from previous infections or vaccinations against specific variants. It can also help identify any potential differences in clinical manifestations, laboratory parameters, and disease severity associated with different variants. By tracking and analyzing a larger number of cases, researchers can gather more robust data that can contribute to evidence-based decision-making in public health measures, such as vaccination strategies and containment measures.

-Your point has been added. Thank you it is completed.

Continuous monitoring of reinfection cases involving a larger number of people is crucial due to the ever-evolving variations of SARS-CoV-2. Conducting larger-scale surveillance can provide researchers with a better understanding of reinfection dynamics and its relationship with different virus variants. Monitoring more reinfection cases can also yield insights into the efficacy of immunity from reinfections or vaccines against specific virus variants. Therefore, it is necessary to analyze more cases to identify potential differences in clinical presentation, laboratory values, and disease severity between the various strains. This will provide us with comprehensive and reliable data that can inform evidence-based decisions about public health interventions, including vaccination campaigns and containment strategies.

Comments on the Quality of English Language

Minor editing

This manuscript is a resubmission of an earlier submission. The following is a list of the peer review reports and author responses from that submission.

Round 1

Reviewer 1 Report

Comments and Suggestions for Authors

The manuscript provides an evaluation on SARS-CoV-2 suspected reinfection in patients screened at the Virology Lab of Istanbul University. The paper needs to be better structured to allow the readers a better comprehension of the retrospective study conducted. Even though SARS-CoV-2 reinfection was a an important topic of debate and research for the scientific community, the present study does not add much to what has already been published especially due to the fact that the cases are few and a quit old. There are areas where the paper could be improved by clarity and dept.

Abstract
I would suggest underlining that you are presenting "your" data. I would reoplace "the results" with "these results" or "our results".

Introduction
Line 27: Covid-2019 is not correct. Replace with
Coronavirus disease 2019 (COVID-19)
Line 28: December 2019. Specify.
Lines 34-38: Rephrase, it is not clear.
Line 41: move "caused by severe..." when you first talk about COVID-19
Lines 44-46: Add some information about the vaccines and the vaccination campain.
Lines 50-54: Better specify the aim of the study presented. Where have cases of re-infection been found? Which country? 

Materials and Methods
I would suggest to better organize this section. I would add an "inclusion/exclusion criteria" paragraph. The inclusion criteria are very confusing. Maybe add a table.
Line 71: "bio-speedy was utilized". For what?

Results
Be careful! When you first use an acronym, please write it in full (e.g., line 138 WBC).
Lines 147-152: I suggest not to use the Ct values to refer to viral load. It is not correct, unless you can relate to a control used to normalize your Ct values.
Lines 154-156: “Based on vaccine type”. Please introduce them before.

Discussion
Lines 180-183: Move to the end where you list the limitation of the study.
Lines 235-250: Please add some references.
Lines: 254-259: Add in the limitation the few and quit old sample analyzed.

Conclusions
I would suggest reorganizing this paragraph to make it more clear.

Comments on the Quality of English Language

The English used is generally good but I recommend a proofreading by a native english speaker to improve the clarity and readability of the paper. 

Reviewer 2 Report

Comments and Suggestions for Authors

The authors of this study state that they aim to investigate the re-infection rates and characteristics of SARS-CoV-2 in individuals with positive RT-PCR results for COVD-19.  First of all, perhaps we should note that RT-PCR results are not positive or negative for COVID-19, but for the presence of SARS-CoV-2 RNA in a clinical sample. Secondly, the authors also state some flaws of the study which, in my opinion, are fundamental. Neither genetic variant typing of the detected strains was performed, nor discussion about rates of re0infection with respect to different time periods when correspondingly different genetic variants prevailed in the population. The authors, also, do not discuss their findings with respect to other studies on re-infection rates. Their discussion section is also very long and full of speculations and personal opinions (which may, or may not be correct), without supporting/discussing them with other researchers' work.

The authors must address these very important issues, in my opinion and once they provide and appropriately discuss the aforementioned information, whenever they have gathered all the necessary data, I would personally suggest publication.